# Development of a Flexible Lightweight Hydraulic-Pneumatic Flywheel System for Wind Turbine Rotors

**Laurence Alhrshy [1],\*, Clemens Jauch [1] and Peter Kloft [2]**

[1] Wind Energy Technology Institute, Flensburg University of Applied Sciences, Kanzleistraße 91-93, 24943 Flensburg, Germany; clemens.jauch@hs-flensburg.de

[2] HYDAC Technology GMBH, Justus-von-Liebig Strasse, Werk 2, D-66280 Sulzbach/Saar, Germany; peter.kloft@hydac.com

\* Correspondence: laurence.alhrshy@hs-flensburg.de; Tel.: +49-461-48161-407

**Abstract:** In this paper, the design of a flexible piston accumulator for application in a hydraulic-pneumatic flywheel system in a wind turbine rotor is presented. The flywheel system enables a wind turbine to vary the inertia of its rotor blades to control the power output and, most importantly, to influence the vibratory behaviour of wind turbine components. The method used for designing the flexible accumulator is based on the one hand on test results of a flexible piston accumulator prototype, and on the other hand, on simulation results of a model of a flexible piston accumulator. As a result, a design of flexible piston accumulators for application in the flywheel system is implemented and compared with the design of conventional steel accumulators. Due to the proposed design of the flywheel system, the impact on the mechanical loads of a wind turbine is analysed. The simulation results show that the new design of the piston accumulators causes a lower impact on the mechanical loads of the wind turbine than a previously published design of piston accumulators. It is further shown that the considered wind turbine can take on the flywheel system without the need for reinforcements in the rotor blades.

**Keywords:** flexible accumulator; flywheel; hydraulic-pneumatic; load simulation; mechanical loads; piston accumulator; wind turbine

## 1. Introduction

For many years, a hydraulic-pneumatic flywheel (FW) system in the rotor of a wind turbine (WT) has been developing at the Wind Energy Technology Institute (WETI) of Flensburg University of Applied Sciences. The general idea of the FW system is based on the variation of the inertia of the rotor blades in a WT. This is done by moving a fluid mass between two piston accumulators in the rotor blade (see Figure 1). Kinetic energy is stored or released by varying the inertia of the rotor blades, which allows a certain variation of the electrical power of the WT, independent of the prevailing wind conditions. Additionally, and most importantly, the vibratory behaviour of the WT can be influenced. The physical description of such an energy storage system is introduced in previous work of the authors, and this covers the kinetics [1] and the hydraulics [2].

As most of the conventional hydraulic piston accumulators are made of steel, the initial design of the piston accumulator for the FW system was made of steel, too. This increases the additional stationary masses of the FW system enormously, and thus the total mass of the rotor blade. Moreover, conventional steel accumulators are not designed to bend with the rotor blade bending. Hence, in the initial design of the FW system, the dimensions of the accumulators, and the location where the accumulators are located in the rotor blades, are chosen such that the accumulators are never bent,

even when the rotor blades are bent most [3]. Because of these two disadvantages, conventional steel accumulators are less suitable for installation in the rotor blades of a WT. Consequently, the intention of the research published in this paper is to design a new piston accumulator for the FW system, that is lighter than steel accumulators on the one hand, and which, on the other hand, allows the piston accumulator to be bent whenever the rotor blade is bent. The challenge of bending a piston accumulator is to prevent the piston from getting stuck and to prevent gas leakages across the piston rings. Carbon fibre-reinforced plastic (CFRP) is a material which is lighter and at the same time more bending-flexible than steel. Also, rotor blades of WTs are made of composite material. This offers better options for attaching the CFRP accumulators to the structure of a rotor blade, compared to steel accumulators. One of the biggest unsolved problems related to conventional steel accumulators is that their attachment to the rotor blade structure should allow the rotor blade to bend around the accumulator without inflicting a bending moment on the accumulator. Since this problem could so far not be solved in an economically feasible manner, steel accumulators were abandoned. Consequently, the focus shifted to the design of flexible CFRP accumulators that are allowed to be bent and that can be attached to the rotor blade structure along their entire length.

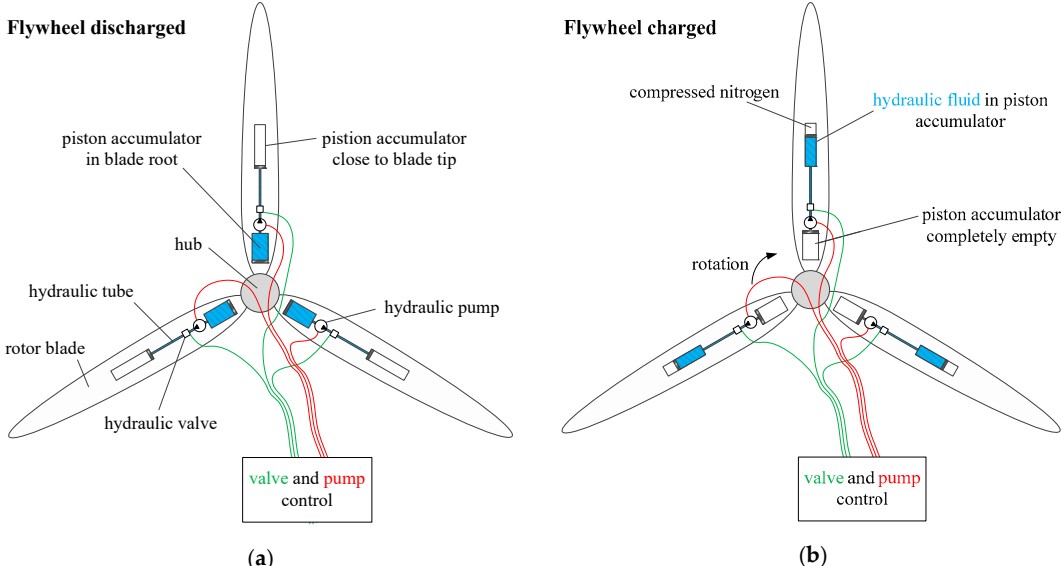

**Figure 1.** Simplified sketch of the flywheel (FW) system: (**a**) flywheel in discharged state and (**b**) flywheel in charged state.

Information about lengths and diameters of the accumulators are obtained from a previous work of the author [3], where a methodology was used to determine the dimensions and positions of the steel accumulators. As a result, several FW configurations were derived. Using the same methodology to derive the FW configurations of CFRP accumulators brings about the advantage that the calculated loads and masses of the CFRP accumulator can be compared with the loads and masses that resulted from the previously investigated steel accumulators.

However, since the goal of this paper is not to compare all the FW configurations published in Hippel et al. [3] for the conventional and for the flexible CFRP design of piston accumulators, the comparison is limited to three relevant FW configurations. The three selected FW configurations are those which represent the largest, a medium and the lowest impact on the mechanical loads. In the following, the considered FW configurations are called "types" and are identified with Roman numerals, as introduced in Hippel et al. [3]. FW type I causes the lowest impact on the mechanical loads of the rotor blades, although this type is the heaviest, and the FW type IV causes the largest impact on the mechanical loads of the rotor blades, although this type is the lightest. FW type VI causes a medium impact on the mechanical loads of the rotor blade compared to the FW types I and IV.

In addition, type VI has an average weight between FW type I and FW type IV. Additionally, these three FW types cover the most relevant locations for installation of piston accumulators inside the rotor blades.

Due to the frequent use of the 5 megawatt (MW) WT of the National Renewable Energy Laboratory (NREL) (Denver, CO, USA) and the public availability of its data [4], the FW system was adapted to this WT. The impact of the FW system on the mechanical loads of the 5 MW WT is analysed by performing load simulations using FAST v7 (Denver, CO, USA). FAST is a computer-aided-engineering tool, which is developed by NREL for simulating the coupled dynamic response of WTs [5]. This load simulation tool, as well as all other state-of-the-art load simulation tools for WTs, is incapable of representing variable rotor blade inertias as they occur in the FW system. Hence, in this work, the load simulation can only be done for two conditions: FW statically charged and FW statically discharged (see Figure 1).

In order to have reference values for the load comparison, load simulation results for the original NREL 5 MW WT rotor blade (without FW system) and for the NREL 5 MW WT rotor blade with steel accumulators are taken from Hippel et al. [6]. These reference values are compared with load simulation results of the NREL 5 MW WT rotor blade with CFRP accumulators. This comparison illustrates the impact of different FW designs on different components of the WT. Based on the results of the load analysis, ultimate and fatigue damages on the rotor blade of the WT can be evaluated, and countermeasures such as reinforcement of the blade can be derived. With this methodology, the solution for the aforementioned problem that steel accumulators had to be abandoned, can be validated.

## 2. Bending Inflicted on Piston Accumulators Due to Rotor Blade Bending

As mentioned above, piston accumulators for FW systems in WT rotor blades should be flexible, i.e., piston accumulators should be connected to the rotor blade along their entire length, which means that they will be bent whenever the rotor blade is bent. For the piston accumulators, this implies that the piston rings have to seal the gap between the piston and the cylinder, even if the cylinder is bent. Also, the piston must not clamp, i.e., it must be free to move under any realistically conceivable bending.

In order to design such a flexible piston accumulator for FW systems in rotor blades of WTs, it is important to know how the considered rotor blade deforms during operation. Hence, simulations of WT operation need to be performed to illustrate the displacement between blade nodes, where a piston accumulator could be installed. Based on the information that is obtained from such simulations, a flexible piston accumulator prototype was built, and tests are performed on this prototype, inter alia, to ensure gas-tightness during bending.

The structural design of the blade applied in the study that is presented in this paper is identical to the Sandia 61.5 m rotor blade [7]. This rotor blade is also applied in the original design of the 5 MW WT developed by NREL [4]. This conception takes advantage of the extensively used 5 MW reference WT in studies by the wind energy research community, as a system that represents the current and future state-of-the-art in offshore systems.

A rotor blade can be bent in edgewise direction and in flapwise direction. In flapwise direction, it is much more flexible than in edgewise direction; hence, it can bend a lot more in flapwise direction. Since a rotor blade can be considered a beam, it can vibrate with different eigenfrequencies, i.e., it has different eigenmodes. Vibration with a particular eigenfrequency leads to deformations of the blade with the respective mode shape. Due to the fact that the first and the second eigenmodes lead to the largest deflections of rotor blades, this study focusses on the first and the second eigenmodes in flapwise direction only. In order to compute the eigenmodes of a beam, like the Sandia 61.5 m rotor blade [7], a finite-element method (FEM) program is needed that provides the coupled modes of the WT blade. Such a program is also provided by NREL and is called BModes [8]. BModes applies the Euler Bernoulli beam theory combined with Hamilton's principle and is capable of computing the coupled modes of beams in general; hence, it can be used for both the blades and the tower of a WT. All the specifications that BModes needs, like rotor speed, blade geometry, and pitch control, can be

derived from the specification of the NREL 5 MW WT [4]. BModes also requires structural property distributions along the rotor blade. These properties are derived from the Sandia 61.5 m rotor blade design [7]. With the aforementioned information, an input file for BModes is prepared, in order to compute mode shapes and frequencies. BModes uses FEM followed by an eigen analysis to compute the eigenmodes [8]. As a result, BModes generates an output file, where the computed coupled mode shapes are shown as spanwise distribution of flap, edgewise, twist, and axial displacement components in a particular mode. Flap displacement is the displacement that the blade exhibits out of the rotor plane, relative to its rest position, when a pitch angle of zero degrees is considered [8]. This component is used to illustrate the nodal displacement for the first and the second eigenmodes in Figures 2 and 3, respectively. Based on this component, slope and curvature of modal displacement curve can be derived as follows, and shown in Figures 2 and 3.

$$slope = \frac{d\ flap\_disp}{d\ blade\_span} \tag{1}$$

$$curvature = \frac{d^2\ flap\_disp}{d\ blade\_span^2} \tag{2}$$

where *blade span* is the span location of a point on the blade reference axis measured with respect to blade root, *flap disp* is the blade displacement out of the rotor plane obtained from BModes, *slope* is the first derivative of the flap displacement with respect to the *blade span*, and *curvature* is the second derivative of the flap displacement with respect to the *blade span*.

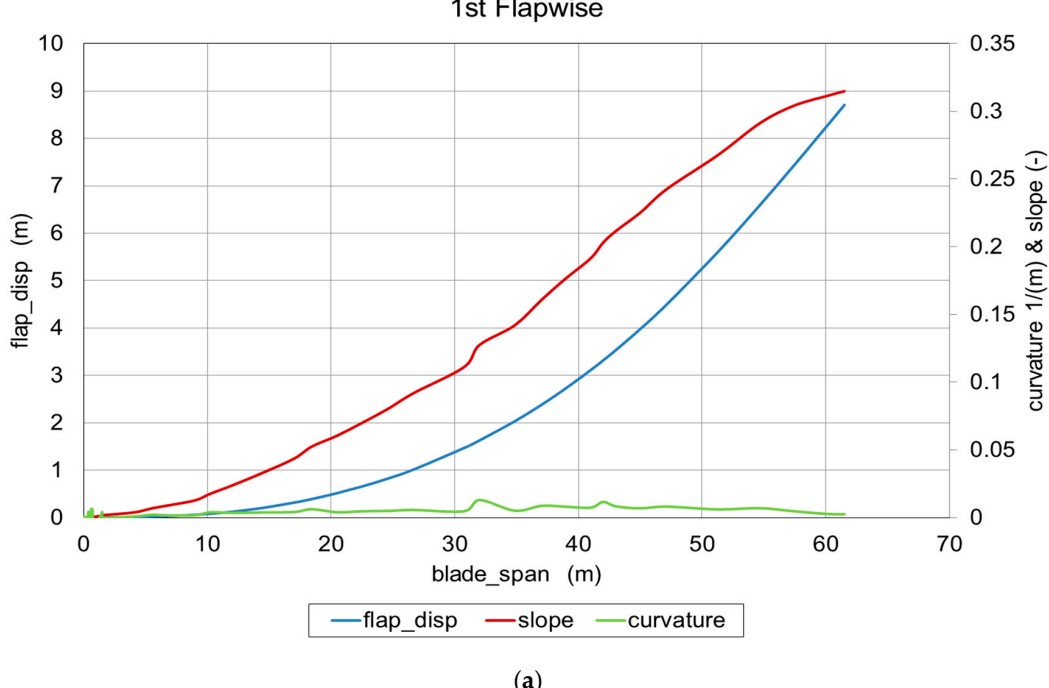

(**a**)

**Figure 2.** *Cont.*

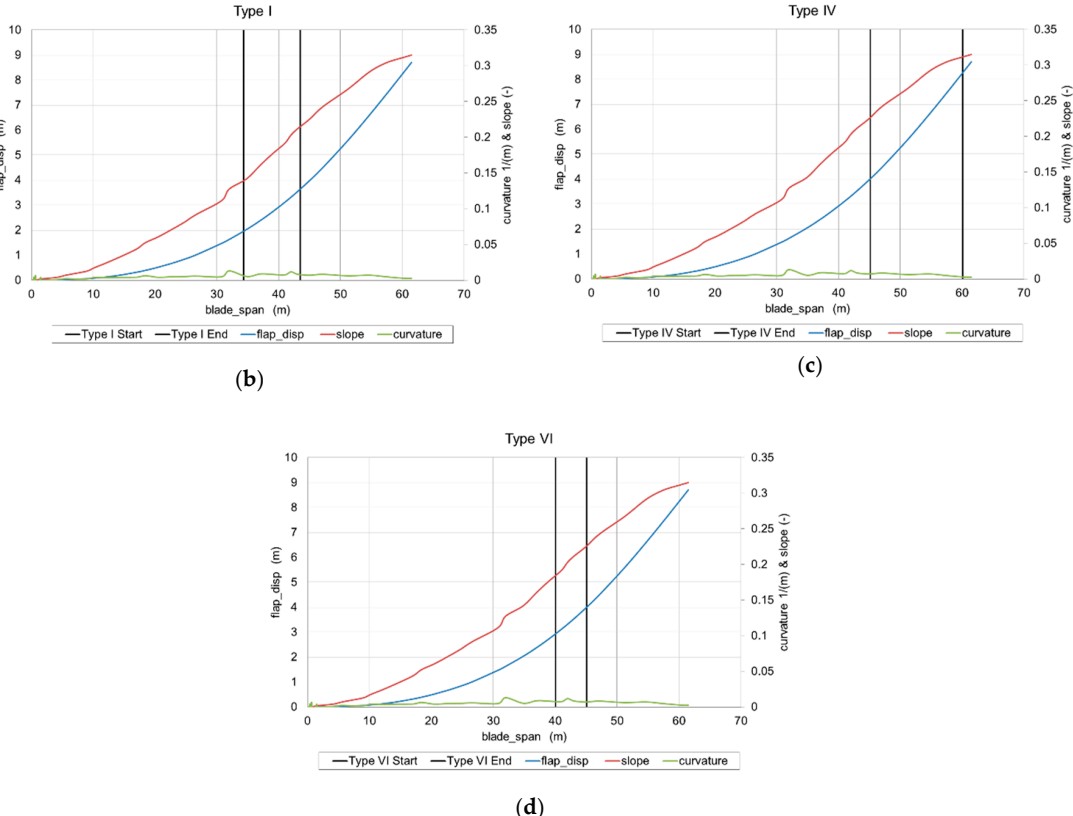

**Figure 2.** (**a**) Flap deflection, slope, and curvature of the first eigenmode of the considered wind turbine (WT) blade, (**b**) location of tip accumulator of type I, (**c**) location of tip accumulators of type IV, and (**d**) location of tip accumulators of type VI.

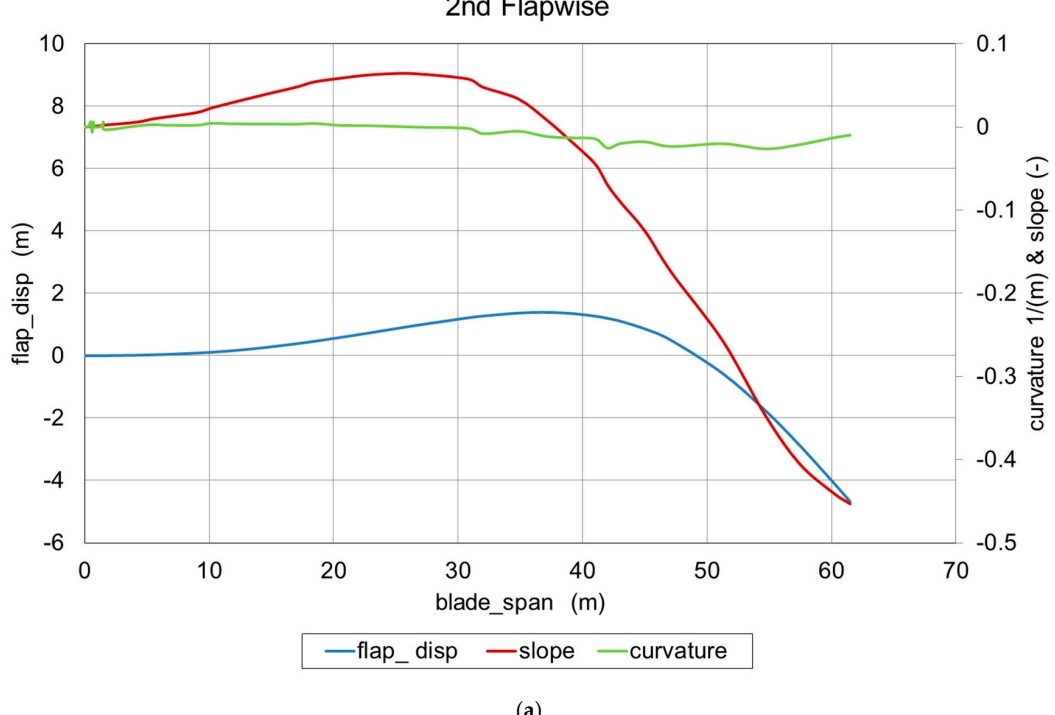

**Figure 3.** *Cont.*

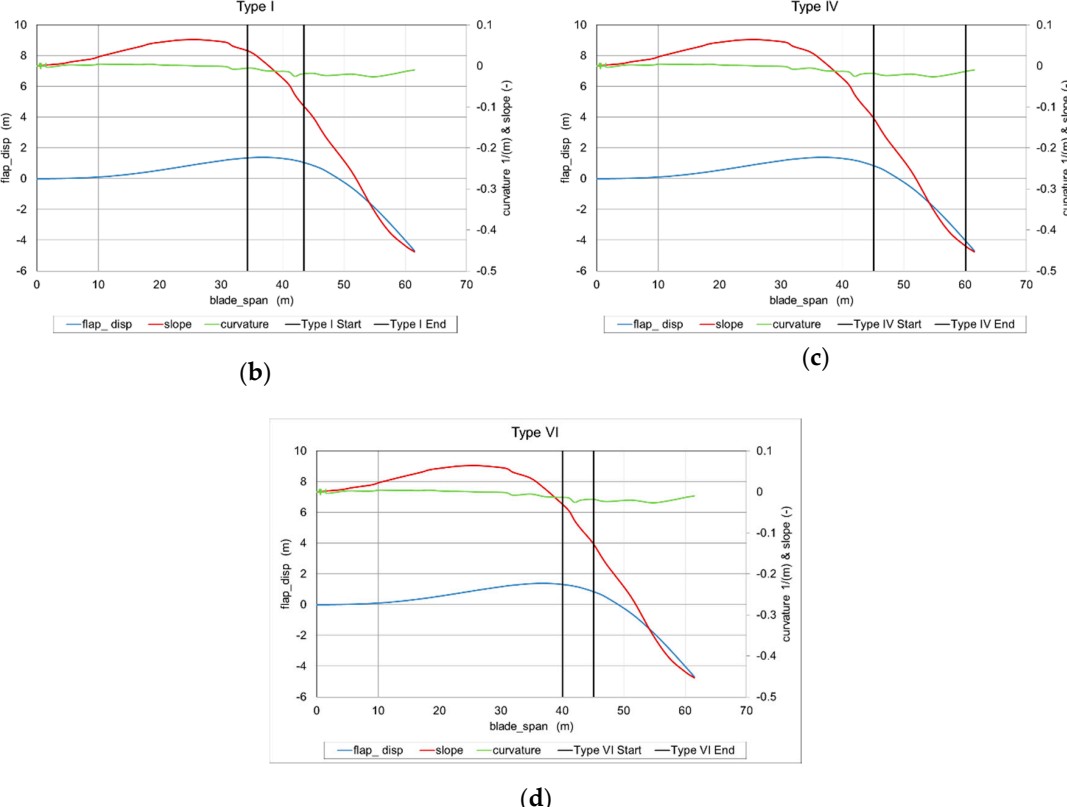

**Figure 3.** (**a**) Flap deflection, slope, and curvature of the second eigenmode of the considered WT blade, (**b**) location of tip accumulator of type I, (**c**) location of tip accumulators of type IV, and (**d**) location of tip accumulators of type VI.

Regarding the three FW types mentioned in the introduction, Figures 2 and 3 show the start and the end location of the tip accumulators, i.e., the tip accumulators are located between the span locations indicated by the vertical black lines in the diagrams (b), (c), and (d) of Figures 2 and 3. For better comprehensibility, the locations of the accumulators are visualized in the different blade sections in Figure 4. Information about flap displacement, slope, and curvature for the installation space of tip accumulators can be derived from Figures 2 and 3. This information can be used for defining the bending test for the flexible accumulator prototype. As a result, the design specifications of the prototype are applied to the FW types. In other words, the design of the CFRP piston accumulators of different lengths, diameters, and pressures, as needed for the three different FW types, are extrapolations of the design specifications of the prototype. This is outlined further in the next section.

Figure 4 presents the available spaces at each rotor blade section (in blue). Where a shows the root accumulator, the fluid pipe, and the tip accumulator (in red) of FW type I inside the rotor blade, Figure 4b, c show those components for FW type IV and FW type VI, respectively.

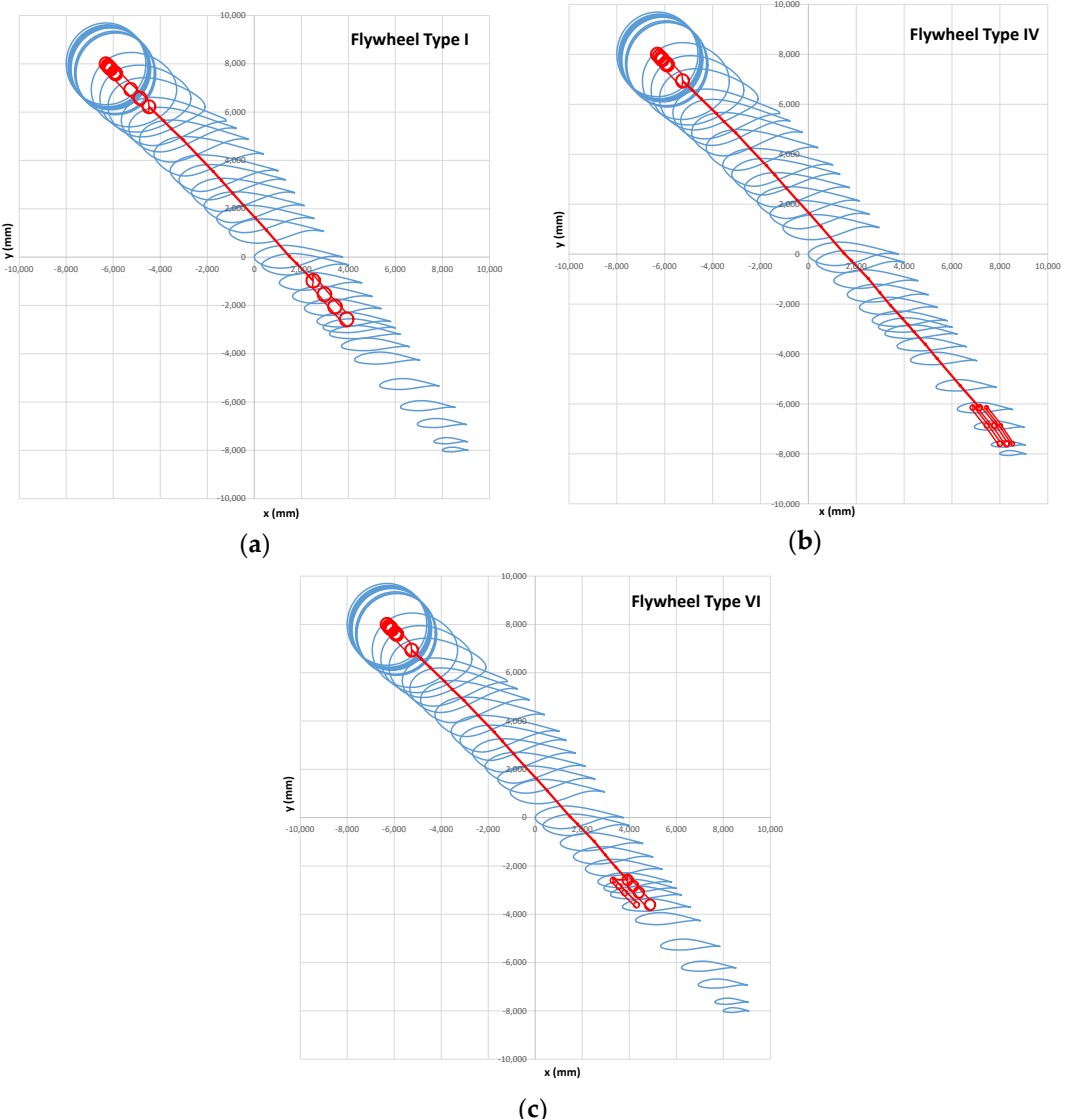

**Figure 4.** Cross-sections of the Sandia 61.5 m blade at the 38 blade nodes and the piston accumulators with fluid pipe of (**a**) FW type I, (**b**) FW type IV, and (**c**) FW type VI.

## 3. Specification of Flexible Carbon Fibre-Reinforced Plastics Piston Accumulator

The work documented in Section 2 leads to the conclusion that the piston accumulators should be used flexibly. They should be connected to the rotor blade along their entire length, which means that they will be bent whenever the rotor blade is bent.

### 3.1. Bending of Conventional Piston Accumulators

It is state-of-the-art that hydraulic-pneumatic piston accumulators must not be bent. Otherwise, it is to be expected that the piston gets stuck and that there are gas leakages from the gas chamber into the fluid chamber or fluid leakages from the fluid chamber into the gas chamber.

Figure 5 shows a bent piston accumulator with conventional piston in the side view. It can be seen that the gap between the piston and the cylinder is the only tolerance that is available for accommodating bending of the cylinder. The drawing in Figure 5 is not to scale; instead, the available gap is strongly exaggerated for illustration. Typically, the gap between piston and cylinder amounts to about 0.15 mm.

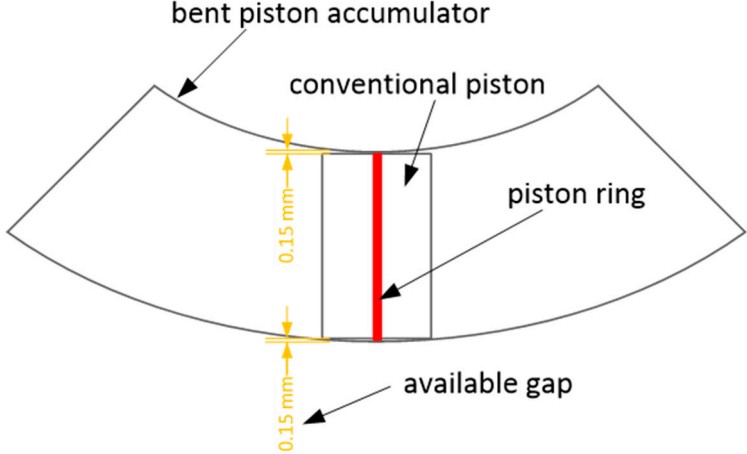

**Figure 5.** Illustration of a bent piston accumulator in the side view (not to scale).

The length of a conventional piston is typically about 60% of the diameter of the piston, as it has to be chosen such that the piston does not tilt under any circumstance. Figure 5 illustrates that the cylinder can only be bent until the gap is fully used up along the length of the piston. If the cylinder is bent further, the piston gets stuck.

Figure 5 shows that the piston ring is fully flattened on the inner radius of the bent cylinder. If the piston ring is not specifically designed for such deformation, it gets damaged. Subsequently, the piston ring would no longer be able to reliably separate gas from fluid, even if the cylinder were no longer bent.

Another problem that leads to leakages from bending the cylinder is the cross-sectional shape of the cylinder, which has to be circular, because the cross-sectional shape of the piston is also circular. However, if a cylinder, which is made of an isotropic material, is bent in the longitudinal direction, its cross-sectional shape becomes an ellipse, see Figure 6.

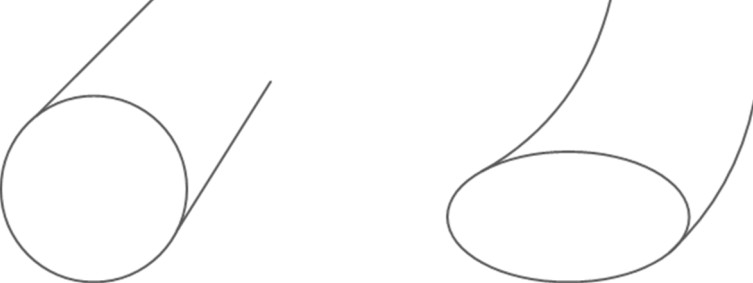

**Figure 6.** Cylinder made of an isotropic material. Left: in resting position with circular cross-section. Right: bent in the longitudinal direction, which turns the cross-section into an ellipse.

In a piston accumulator, the piston is usually a lot more rigid than the cylinder. Therefore, it can neither bend, nor can it match its cross-sectional shape to the cross-sectional shape of the cylinder. Consequently, there will be leakages between the circular piston and the elliptic cylinder.

*3.2. Preliminary Design of Flexible Piston Accumulators*

The problems mentioned above are solved by HYDAC Technology GmbH (HYDAC.com, Sulzbach, Saarland, Germany), who design flexible piston accumulators made of CFRP. This means that the piston rings have to seal the gap between the piston and the cylinder, even if the cylinder is bent. Also, the piston must be free to move under any bending. The material CFRP is advantageous because it is anisotropic, and hence, allows designing different properties in different directions. Furthermore, use of CFRP should lead to a weight reduction compared to conventional steel piston accumulators.

This motivated HYDAC, in cooperation with PRONEXOS (pronexos.com, Almelo, The Netherlands), to perform a test to specify a flexible CFRP piston accumulator with a metal liner on the inside of the CFRP cylinder.

In the process of manufacturing the cylinder, the anisotropic properties can be achieved by winding the carbon fibres in certain directions around the cylinder. Figure 7 qualitatively shows the different winding directions.

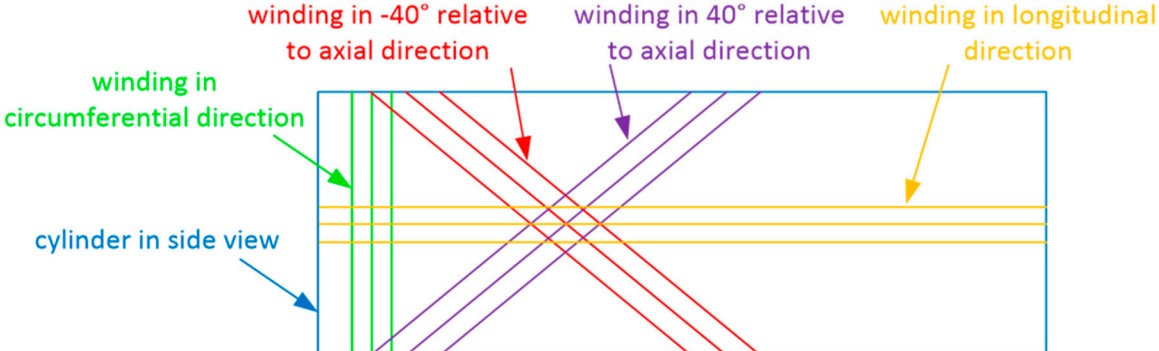

**Figure 7.** Qualitative illustration of the different winding directions that can be applied in the manufacturing process of a carbon fibre-reinforced plastic (CFRP) cylinder (the angles are only examples for illustrating the concept).

These winding directions lead to the following properties of the laminate:

- Windings in the circumferential direction increase the stiffness in the radial direction. The more carbon fibre layers are added in this direction, the less of a diameter increase the cylinder exhibits in case of internal pressure. Also, the cross-sectional shape is less prone to become an ellipse in case of longitudinal bending.
- Windings in the longitudinal direction increase the stiffness in the axial direction. By adding a specific number of windings in the longitudinal direction, the cylinder deflects by a certain extent from a specific external force in the lateral direction. The number of windings in the longitudinal direction has to be chosen such that these fibres rupture when the bursting pressure inside the cylinder is exceeded.
- Windings in a certain angle (positive and negative), with respect to the longitudinal axis, add to stiffness in the radial direction and in the axial direction, to an extent that depends on the angle. Most importantly, these windings add to the torsional stiffness of the cylinder (the angles shown in Figure 7 are only arbitrary examples for illustration).

When a piston accumulator is to be bent, the jamming of the piston, as shown in Figure 5, has to be prevented. At the same time, the piston has to reliably separate gas from fluid, i.e., the piston rings must not get damaged from bending the cylinder. These properties are achieved by making the piston from three flexibly connected discs, see Figure 8.

The outer discs (called stabilising discs in Figure 8) prevent the piston from tilting. The centre disc carries the piston ring, and therefore, separates the gas chamber from the fluid chamber. The three discs are connected to each other via a flexible rod, which can be bent easily when the cylinder around the piston gets bent.

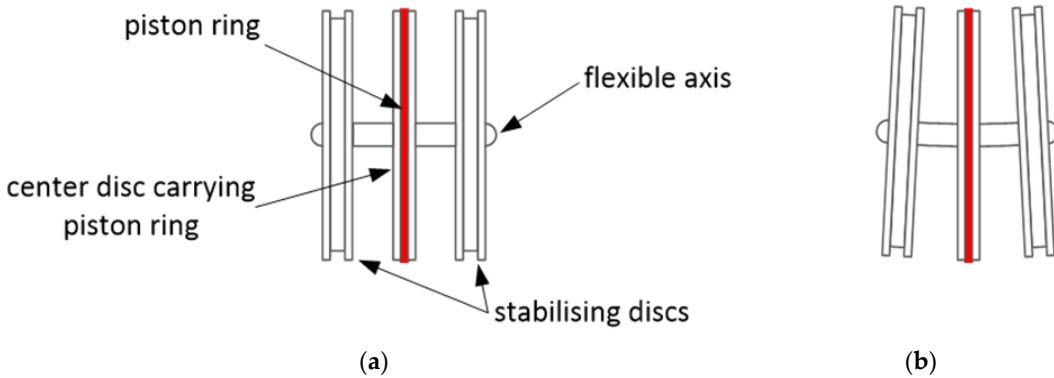

(**a**)  (**b**)

**Figure 8.** Flexible piston (**a**) with naming of components and (**b**) in bent state.

*3.3. Test Specifications*

Based on the information about the bending behaviour of the Sandia 61.5 m rotor blade presented in Section 2, HYDAC, in cooperation with PRONEXOS, built a prototype of a CFRP piston accumulator with a steel liner on the inner surface of the CFRP tube, see Figure 9. Figure 9 shows the steel liner covered with the CFRP laminate. In the background, the piston can also be seen. This prototype is tested in the test area of HYDAC Technology. Further design specifications for the prototype are:

- The structural strength of the laminate with the steel liner shall withstand the working pressure of 60 bar.
- A maximum diameter increase of the accumulator tube of 0.1 mm at 40 bar working pressure.
- The laminate should rupture in the longitudinal direction at burst pressure (>120 bar).
- The CFRP design shall be no more than 1.5 times as expensive as a comparable steel design.
- The steel liner at the inner surface shall have a wall thickness of 1 mm and its main purpose is to guarantee gas leak tightness, even in case of inter-fibre cracks in the CFRP laminate.

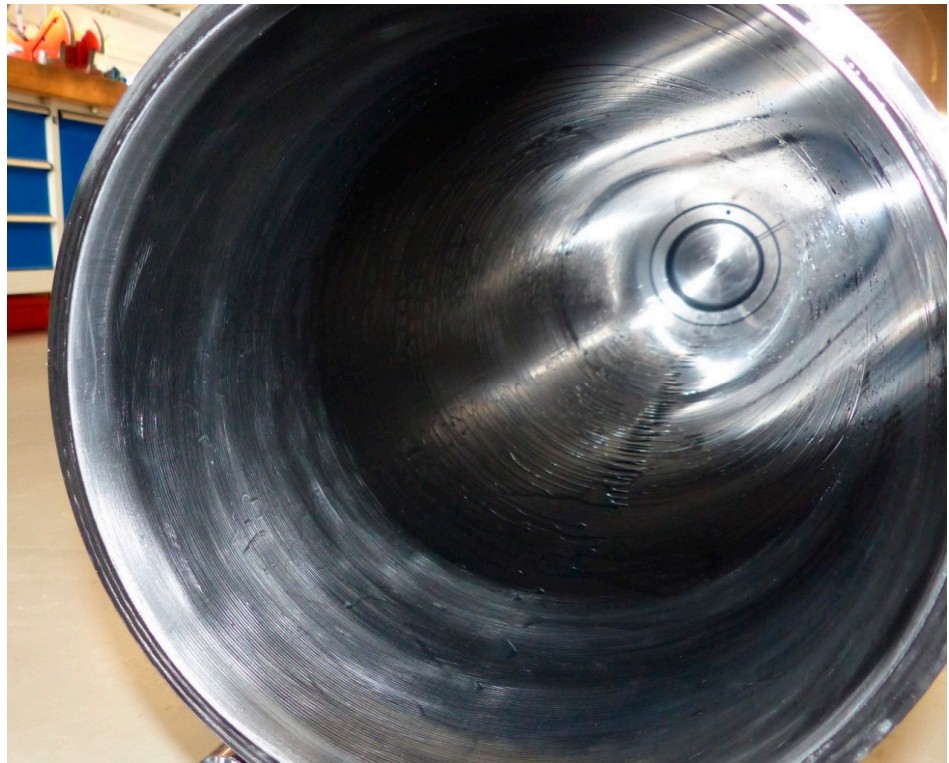

**Figure 9.** A prototype of a CFRP piston accumulator with a steel liner on the inner surface of the CFRP tube.

The materials used for the construction of the prototype are already qualified for series manufacturing at PRONEXOS. The steel liner of the prototype is produced via a cold-forming technology. Table 1 illustrates the material properties and the lay-up plan of the CFRP prototype.

**Table 1.** Laminate lay-up and the material properties of the CFRP with steel liner prototype.

| Lay-Up No. | Thickness mm | Direction ° | Density g/cm³ | $E_{\parallel}$ MPa | $E_{\perp}$ MPa | $G_{\perp\parallel}$ MPa | $v_{\perp\parallel}$ - |
|---|---|---|---|---|---|---|---|
| 1 | 0.83 | 88 | 1.57 | 146,342 | 10,725 | 5376 | 0.2555 |
| 2 | 1.06 | 15 | 1.57 | 146,342 | 10,725 | 5376 | 0.2555 |
| 3 | 0.83 | 88 | 1.57 | 146,342 | 10,725 | 5376 | 0.2555 |
| 4 | 0.83 | 88 | 1.57 | 146,342 | 10,725 | 5376 | 0.2555 |
| 5 | 1.06 | 15 | 1.57 | 146,342 | 10,725 | 5376 | 0.2555 |
| 6 | 0.83 | 88 | 1.57 | 146,342 | 10,725 | 5376 | 0.2555 |
| 7 | 0.83 | 88 | 1.57 | 146,342 | 10,725 | 5376 | 0.2555 |
| 8 | 1.06 | 15 | 1.57 | 146,342 | 10,725 | 5376 | 0.2555 |
| 9 | 0.83 | 88 | 1.57 | 146,342 | 10,725 | 5376 | 0.2555 |
| 10 | 1 | - | 7.80 | 210,000 | 210,000 | 79,300 | 0.3200 |
| **Total thickness** | 9.16 | - | | | | | |

Where $E_{\parallel}$, $E_{\perp}$, $G_{\perp\parallel}$, and $v_{\perp\parallel}$ are the Young's modules parallel to fibre direction, the Young's modules perpendicular to fibre direction, the in-plane share modules, and the major Poisson's ratio of a CFRP laminate layer, respectively.

In order to verify the validity of the prototype specification, the prototype is modelled with the design tool COMPOSITOR v4.1 [9]. COMPOSITOR is a structural computer-aided-engineering tool, which was developed by the Institute for Plastics Processing at RWTH Aachen University (ikv-aachen.de, Aachen, Nordrhein-Westfalen, Germany).

For the working pressure of 40 bar, a load case of two global stresses can be translated on the basis of Barlow's formula [10]:

$$\sigma_t = \frac{p \cdot d_m}{2 \cdot s} \tag{3}$$

$$\sigma_a = \frac{p \cdot d_m}{4 \cdot s} \tag{4}$$

where $\sigma_t$ and $\sigma_a$ are the circumference and the longitudinal stresses, respectively. $p$ is the internal pressure, $d_m$ is the mean diameter, and $s$ is the wall thickness of the pressure vessel, here, the tube of the piston accumulator. The material specifications and the laminate lay-up of the flexible accumulator prototype are considered in the simulation model of COMPOSITOR. As a result, two components of the strain, in circumference and longitudinal directions, are computed, based on the classical laminate theory. The resulting circumference strain can be transformed into a diameter increase, i.e., the diameter increase of the test model is about 0.08 mm at the working pressure of 40 bar, which is lower than the required maximum diameter increase of 0.1 mm.

The dimensions of the flexible piston accumulator prototype are listed in Table 2. These dimensions are limited by the dimensions of the test facility at HYDAC's test area. Hence, the prototype size is obviously not comparable with the dimensions of a piston accumulator for a FW system in a multi MW WT rotor blade.

**Table 2.** Flexible piston accumulator prototype dimensions.

| Inner Diameter, mm | Outer Diameter, mm | Total Length, mm | Stationary Weight, kg |
|---|---|---|---|
| 250 | 269.2 | 1995 | 89.78 |

The test requirement of maximum diameter increase of 0.1 mm is so conservative that it can only be achieved by relatively small piston accumulators. Applying this requirement to the FW types for a multi MW WT leads to oversizing of the wall thickness of the piston accumulators. Therefore, the maximum diameter increase is slackened to 0.3 mm, which is a commonly used value for steel accumulators. A larger diameter increase poses a challenge for the piston rings, as they must not extrude into the gap between liner and piston. However, a 0.3 mm diameter increase is a feasible value that is confirmed by piston accumulators that have been operating for a long time.

## 4. Carbon Fibre-Reinforced Plastic Flywheel Types

### 4.1. Geometry

The possible installation spaces for the tip accumulators are limited by the dimensions of the airfoils of the Sandia 61.5 m blade design. In this blade, like in all state-of-the-art rotor blades, the space inside the blade decreases towards the blade tip. As a result, a single, double, or triple tip accumulator can be installed inside the blade, in order to achieve the desired increase of the inertia of the NREL 5 MW WT.

The properties of the CFRP tip accumulators of the three selected FW types are summarised in Table 3, where rotor radius is the maximum rotor radius that can be achieved by the tip accumulators. This data is further needed in the mass and load comparison in Sections 4.2 and 5.

**Table 3.** Properties of the CFRP piston accumulators of the three FW types.

| Flywheel Types | No. of Tip Accumulators | Rotor Radius, m | Inner Diameter, mm | Total Length, mm |
|---|---|---|---|---|
| Type I | 1 | 40.01 | 577 | 5477 |
| Type IV | 3 | Each at 61.6 | 224, 209, and 149 | 5966, 5957, and 5920 |
| Type VI | 2 | Each at 46.6 | 443 and 241 | 5047 and 4923 |

Based on the information about the CFRP tip accumulators' properties mentioned in Table 3, the proposed installation spaces inside the Sadia 61.5 m rotor blade are illustrated in Figure 4.

### 4.2. Masses Comparison

In order to design the CFRP piston accumulators for the different FW types, according to the prototype specifications (see Table 1), again, COMPOSITOR is applied. For this purpose, the model specifications of the prototype, as introduced in Section 3.3, are used to model the CFRP accumulator as needed for the three FW types. The three FW types that are mentioned in the Introduction Section are analysed, and the wall thicknesses of the CFRP accumulators are computed according to the working pressure of each FW type and to the maximum permissible diameter increase of the steel accumulator, as discussed above.

As a result, simulations of the CFRP piston accumulators for the FW systems in COMPOSITOR have shown that in addition to the flexibility of the CFRP piston accumulator, also, the total stationary mass is decreased compared to conventional steel piston accumulators. This is due to the special technology used by winding the carbon fibre in circumferential, longitudinal, and hoop and helical (e.g., ±40° relative to axial direction) directions around the steel liner of the piston accumulator, see Figure 7.

Consequently, the design of the flexible piston accumulator is noticeably lighter than the conventional steel design. Table 4 and Figure 10 compare the properties and the masses of both CFRP and conventional steel piston accumulators for the three FW types mentioned in the Introduction Section. In Figure 10, the CFRP accumulators and the conventional steel accumulators are represented as CFRP + Liner and as Steel in blue and red, respectively. The masses represented in Figure 10 are the total stationary masses of root and tip accumulators of each FW type, where type I has only one tip

accumulator. Type IV and type VI have three and two tip accumulators, respectively. Figure 10 shows the obvious weight reduction of the total stationary masses of the root and the tip accumulators for the CFRP design compared to the conventional steel design for all represented FW types. The effect of this weight reduction on the impact of the mechanical loads of the WT is discussed in the next section.

**Table 4.** Properties of the CFRP and the conventional piston accumulators of the three FW types.

| Flywheel Types | CFRP + Liner Accumulators | | | | | | Steel Accumulators | | | | | |
|---|---|---|---|---|---|---|---|---|---|---|---|---|
| | Type I | Type IV | | | Type VI | | Type I | Type IV | | | Type VI | |
| No. of tip accumulators | 1 | 1 | 2 | 3 | 1 | 2 | 1 | 1 | 2 | 3 | 1 | 2 |
| Wall thickness mm | 4.24 | Each 3.72 | | | Each 3.72 | | 3 | Each 3 | | | Each 3 | |
| Working pressure bar | 13.9 | Each 31.6 | | | Each 18.7 | | 13.9 | Each 31.6 | | | Each 18.7 | |
| Material | carbon fibre Toho tenax STS40 and→S335 steel liner | | | | | | 34CrMo4 steel | | | | | |

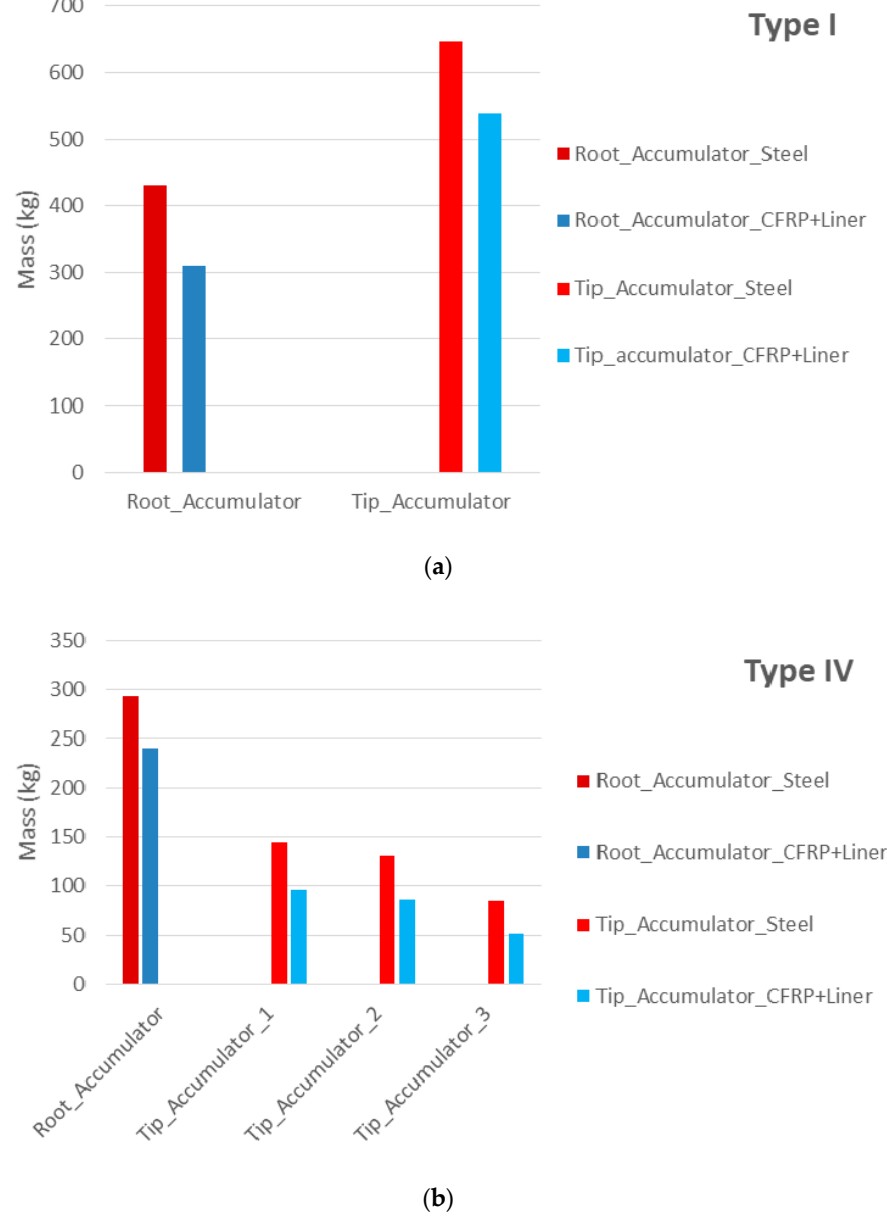

(**a**)

(**b**)

**Figure 10.** *Cont.*

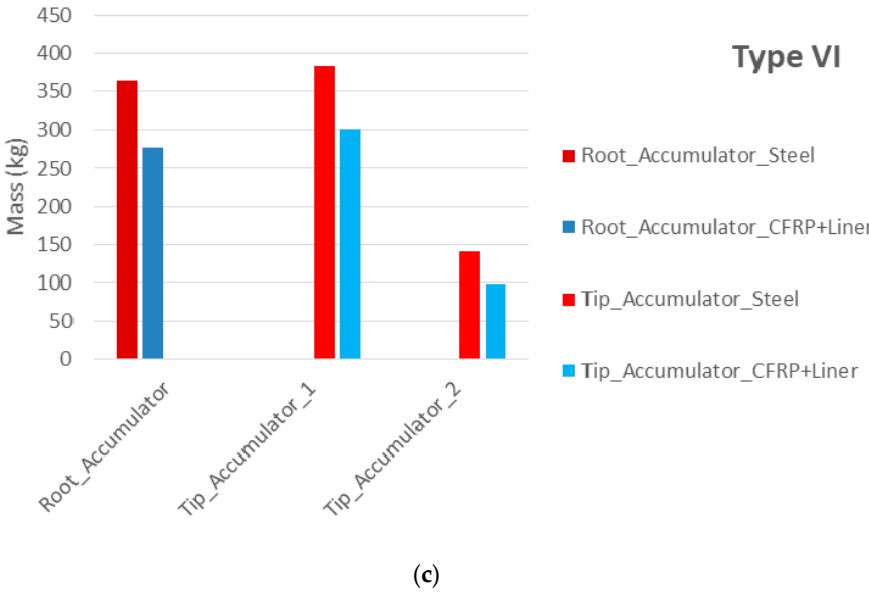

(**c**)

**Figure 10.** Mass comparison of blade tip and blade root accumulators from steel and from CFRP with steel liner for (**a**) type I, (**b**) type IV, and (**c**) type VI.

## 5. Load Analysis

The load analysis aims to quantify the effect of the weight reduction of the CFRP piston accumulators on the impact of the mechanical loads of the NREL 5 MW WT. The quantifying of loads is done firstly by computing the mechanical loads of the original WT without a FW system in order to get reference load values. Afterwards, load simulations are done for the WT with a FW system comprising CFRP accumulators as well as with a FW system comprising conventional steel accumulators in the rotor blades. Eventually, the deviations of the mechanical loads from the reference values illustrate the impact of the different FW system designs on the mechanical loads of the WT.

For this analysis, the program FAST is used. FAST is an aeroelastic computer-aided-engineering tool developed by NREL [4], that allows for computing mechanical loads in a wind turbine.

As described in previous work of the author [6], the methodology used to implement the variable behaviour of the FW system in FAST considers the FW system statically, i.e., either the system is fully charged or fully discharged. In state-of-the-art WTs, the rotor blade inertia does not vary during operation. Therefore, all common load simulation tools, like Felx5 (http://www.dtu.dk, v5, Technical University of Denmark, Kgs. Lyngby, Denmark), FAST (https://nwtc.nrel.gov/FAST8/), or Bladed [11], are unsuitable for simulating the dynamic operation of the FW system. This would require drastic adaptations of the source code of such load simulation tools. Future work will focus on implementing variable blade inertias in a load simulation tool to allow for the simulation of the dynamic operation of a FW system in a WT rotor.

The structural blade properties of the Sandia 61.5 m rotor blade design are described in FAST as sectional distributed mass densities and stiffnesses. Each section with its mass and stiffness is at a particular location in the spanwise direction of the blade. These locations are also the locations where cross-sections of the airfoil are illustrated in Figure 4. As the eigenmodes characterise the structural properties of the rotor blade, FAST requires also the mode shape coefficients of the first three eigenmodes. The impact of the FW system on the structural properties of the rotor blade is implemented in terms of increasing the mass density at those nodes of the blade, where the root and the tip accumulators are installed. Regarding the state of the charge of the FW system, the fluid mass is shifted between the tip accumulator and the root accumulator, i.e., if the FW system is charged, the complete fluid mass is added to the mass density of the nodes of the blade tip, where the tip accumulator is located. If the FW system is discharged, the complete fluid mass is added to the mass

density of the nodes of the blade root where the root accumulator is located. These two scenarios are done for the three previously mentioned FW types. Additionally, the mode shape coefficients of the first three eigenmodes are computed with BMode for both states of charge, due to the change of the blade section masses. Other structural properties of the blade input data remain unchanged.

In this paper, load simulation results of the cases "original WT without FW system" and "WT with conventional steel accumulator FW system" are obtained from previous work of the author [6]. Hence, it is mandatory for the load simulations of the WT with a FW system comprising CFRP accumulators to take the same design load cases (DLCs) and the same wind fields. Only in this way is it possible to compare load simulation results of both FW system designs with the original WT without FW system. The selected DLCs are based on the "IEC International Standard for the Design of Wind Turbines" and they cover the most important operation modes of a WT [12]. Thus, the computed ultimate and fatigue loads of the unchanged rotor blade and of the rotor blade with the conventional steel piston accumulators can be obtained from Hippel et al. [6]. The results of these simulations are used as reference values. Subsequently, load simulations are performed for the WT with FW system comprising the CFRP piston accumulators. Loads are computed for the WT with the FW types I, IV, and VI in charge and discharged state. The resulting ultimate and fatigue loads are compared to the reference values. Afterwards, the changes in ultimate and fatigue loads of rotor blade, tower, and drive train are evaluated, and potentially occurring structural blade failures are discussed.

At first, the change in ultimate loads of the blades, the drive train, and the tower of the 5 MW WT due to both designs of the FW system are discussed. Figure 11 depicts the change in ultimate loads as a percent increase in bending moments. The charged and the discharged states of the FW system with CFRP accumulators are represented in Figure 11 in dark red and dark blue, whereas the charged and discharged states of the FW system with conventional steel accumulators are shown in light red and light blue, respectively. The increase in bending moments when both FW designs are in a charged state are obviously greater than the bending moments when the FWs are in a discharged state. The reason for this is that the fluid mass in the charged state is added to the stationary mass of the tip accumulator, where the lever arm of the gravitation force has a larger distance to the centre of rotation. Also, the larger distance to the centre of rotation leads to larger centrifugal forces created by the fluid mass. Due to the stationary weight reduction of the CFRP accumulator, the increase in bending moments of almost all FW types is lower for the CFRP design compared to the conventional steel design. Excepted from this is the change in bending moment at blade cuts for FW Type IV in the charged state. Only in this scenario is the increase in bending moment of the CFRP design about 2% greater than the conventional steel design. The reason for this minor increase is the difference in WT operating points of both FW designs when the maximum turbulent wind speed hits the rotor blade.

The first and the second graphs in Figure 11 show the bending moments at the blade cuts, *SpnMLxB*, and the tower cuts, *TwHtML*, where the maximum increase of these moments are indicated. The variable names are based on the conventions used in FAST [5].

Finally, the change in fatigue loads are analysed in a similar procedure as for ultimate loads. Fatigue analysis is done twice for each FW design over the lifetime of 20 years of the WT. Once, the FW system is charged for 20 years, and once, again, the FW system is discharged for 20 years. This is, again, due to the fact that the dynamic behaviour of the FW system cannot be implemented in FAST.

Figure 12 compares the change in fatigue loads for both FW designs at the rotor blade, the tower, and the drive train of the 5 MW WT. The light red and the light blue bars in Figure 12 represent the charged and the discharged states of the conventional design of the FW system, respectively. The columns in dark red and in dark blue represent the charged and the discharged states of the CFRP design of the FW system, respectively.

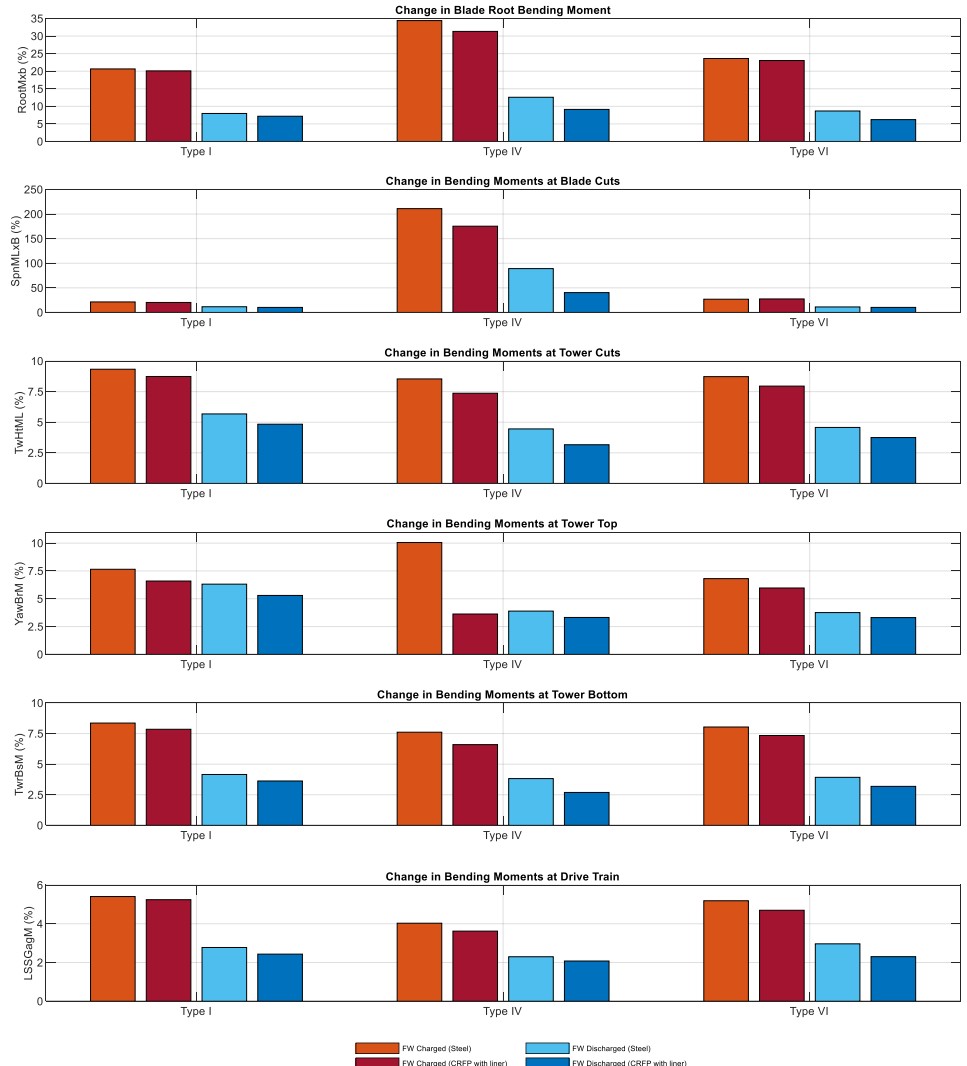

**Figure 11.** Comparison of changes in ultimate loads in (from top to bottom) blade root bending moment, bending moments at blade cuts, bending moments at tower top, bending moments at tower bottom, and bending moments at drive train, between steel accumulator and CFRP with metal liner accumulator for FW types I, IV, and VI in charged (red) and discharged (blue) status, both with respect to the ultimate loads of the original WT without FW system.

The change in fatigue-damage equivalent bending moment at the blade, the tower, and the drive train caused by the CFRP design of the FW system are obviously lower than the change in fatigue-damage equivalent bending moment caused by the conventional steel design. This is due to the difference of the stationary weight between the CFRP design and the conventional steel design of the piston accumulators. For the FW type IV, the fatigue equivalent bending moment at tower top in discharged state and at tower bottom in charged and discharged states are slightly decreased for both FW designs. This slight decrease of the fatigue bending moment can be traced to the low impact of type IV on the tower top and tower bottom during the simulation time of 20 years of operation.

Consequently, ultimate and fatigue failures are evaluated to ensure that there is no need for reinforcement in the original rotor blade design of the NREL 5 MW WT. The failure evaluation in this paper is based on the same methodology used by Hippel et al. [6], where the load simulation results were used to indicate ultimate and fatigue failures. The failure evaluation in a previous work of the author [6] showed that neither ultimate nor fatigue failure of the rotor blade was caused by the presence of the conventional steel accumulator FW system. Figures 11 and 12 show that the loads of

the CFRP FW design are in general lower than those of the conventional steel FW design. This leads to the conclusion that the increase in ultimate and fatigue loads due to the installation of the CFRP FW design cause neither ultimate nor fatigue failure of the rotor blade, since the WT model and the rotor blade design are the same for both load simulations. Hence, with the flexible CFRP accumulators as designed here, and with the methodology for designing a FW system with these accumulators, a FW system can be implemented in a WT rotor. Although the previous work already indicated that a FW system could be installed in a rotor without causing failure in the blades, there was always the shortfall that the conventional steel accumulators must not be bent. Therefore, a mounting technology would have been required that attaches the accumulators to the rotor blades, but that, at the same time, allows the rotor blade to bend around the accumulator without inflicting a bending moment on the accumulator. With the flexible CFRP accumulators, this problem is solved. These accumulators can be laminated to the structure of the rotor blade. Hence, the accumulators bend with the rotor blade, which is no problem for these flexible CFRP accumulators.

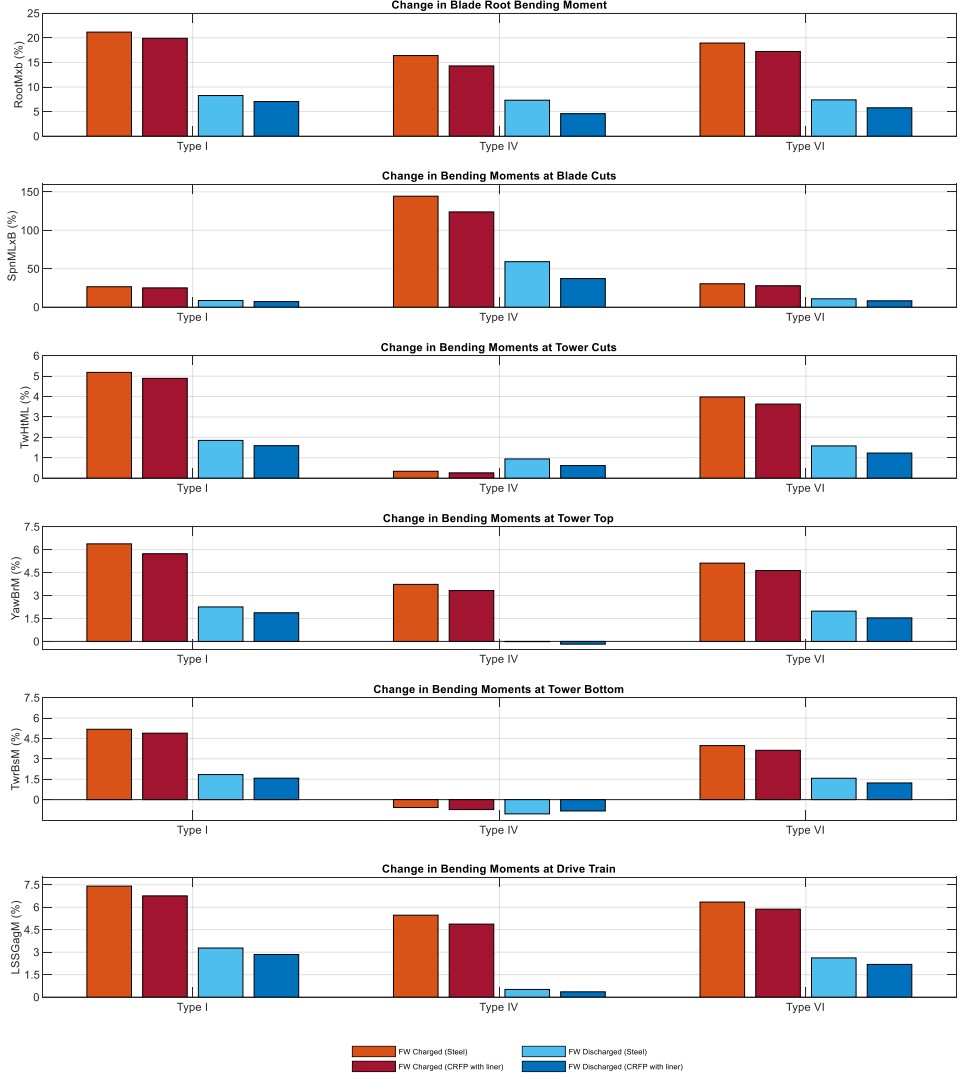

**Figure 12.** Comparison of changes in fatigue loads in (from top to bottom) blade root bending moment, bending moments at blade cuts, bending moments at tower top, bending moments at tower bottom, and bending moments at drive train, between steel accumulator and CFRP with metal liner accumulator for FW types I, IV, and VI in charged (red) and discharged (blue) status, both with respect to the ultimate loads of the original WT without FW system.

## 6. Conclusions

In the present paper, a CFRP piston accumulator design was presented for a FW system to be installed in a rotor blade of a WT. The development of such CFRP accumulators goes through different stages. Firstly, the bending behaviour of the considered rotor blade, here the Sandia 61.5 m rotor blade for the NREL 5 MW WT, was analysed, and the bending gradients were derived for the first and second eigenmodes. Subsequently, based on the derived bending gradient, a flexible CFRP piston accumulator prototype was specified, built, and tested. The design specifications of the prototype were used to derive the different piston accumulators needed for the FW system. Finally, ultimate and fatigue load comparisons were done between a WT with a conventional steel design of FW system, a WT with a flexible CFRP design FW system, and a WT without any FW system. The load comparison revealed that the increase in ultimate and fatigue loads of the blade, the tower, and the drive train due to the CFRP FW design are much smaller than the increase in these loads in case of the conventional steel FW design. Furthermore, the load analyses of the 61.5 m rotor blade revealed that neither ultimate nor fatigue failure occurred due to the installation of the CFRP FW design. Consequently, the initial design of the NREL 5 MW WT can take on the FW system without the need for reinforcement in the rotor blades.

It can be concluded that the main goal of designing a FW system made of piston accumulators that can be realistically implemented in rotor blades has be fully achieved. The mounting technology could not be simpler: the designed CFRP accumulators are allowed to bend with the rotor blade. Also, they are made of epoxy, which allows to laminate them to the glass fibre-reinforced plastic, of which the rotor blade is made.

With the methodology presented in the paper, a FW system can be designed for any arbitrary WT. Although the methodology is presented for the case of the NREL 5 MW reference WT, it can be applied to any other WT, too.

Future work will focus on designing FW types that take full advantage of the fact that the newly developed accumulators may be bent, i.e., the development of very long accumulators will focus on designing FW types where the fluid mass can move through most of the length of the rotor blade. In order to assess the effect of such FW types on the mechanical loads, it becomes even more important to adapt load simulation tools to this purpose. Hence, a load simulation tool that allows simulating variable blade inertias will need to be developed in the future.

**Author Contributions:** Conceptualization, L.A. and C.J.; methodology, L.A. and C.J.; software, L.A.; validation, L.A., C.J. and P.K.; formal analysis, L.A.; investigation, L.A.; resources, C.J. and P.K.; data curation, L.A., and C.J.; writing—original draft preparation, L.A., and C.J.; writing—review and editing, L.A. and C.J; visualization, L.A., C.J. and P.K.; supervision, C.J. and P.K.; project administration, C.J. and P.K.; funding acquisition, C.J. and P.K. All authors have read and agreed to the published version of the manuscript.

**Funding:** This work was supported by the Gesellschaft für Energie und Klimaschutz Schleswig-Holstein GmbH (EKSH); Project Nr. 8/12-29.

**Conflicts of Interest:** The authors declare no conflict of interest.

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
