# Peer review of "Development of a Flexible Lightweight Hydraulic-Pneumatic Flywheel System for Wind Turbine Rotors"

_fluids, doi:10.3390/fluids5040162_

Round 1
Reviewer 1 Report
1.In page 9, Figure 7 seems recurring three times. It should be corrected before publication.
2. This paper propose an useful and novel design of a flexible lightweight hydraulic-pneumatic flywheel system in a wind turbine rotor. I suggest this manuscript is suitable for the publication in the journal of fluids.
Author Response
Dear Reviewer,
We wish to thank you for the time and effort dedicated to provide feedback on our manuscript. We are grateful for you constructive comments, which provide a valuable improvements to our paper. The suggestions you made are carefully incorporated and revised. See below, in black, the reviewers’ comment, and; in blue, our point-by-point response to the reviewers’ comments. All changes within the manuscript are highlighted using the “Track Changes” function in Microsoft Word.
- In page 9, Figure 7 seems recurring three times. It should be corrected before publication.
Thank you very much for catching this confusing error. It was a mistake by using the cross-reference of Figure 7 in line 218. The cross-reference referred to the entire caption of figure 7. We have now corrected this error.
- This paper propose an useful and novel design of a flexible lightweight hydraulic-pneumatic flywheel system in a wind turbine rotor. I suggest this manuscript is suitable for the publication in the journal of fluids.
Thank you, we appreciate that.
Best regards
Reviewer 2 Report
Figure 1b is in my opinion inaccurate and requires correction.
I also have problems with Figure 7. I don't know if this is a manuscript processing error. Please check it
Table 1 and 4 is also split up and it's not looking good.
Figure 11 and 12 has too little descriptions
Author Response
Dear Reviewer,
We wish to thank you for the time and effort dedicated to provide feedback on our manuscript. We are grateful for you constructive comments, which provide a valuable improvements to our paper. The suggestions you made are carefully incorporated and revised. See below, in black, the reviewers’ comment, and; in blue, our point-by-point response to the reviewers’ comments. All changes within the manuscript are highlighted using the “Track Changes” function in Microsoft Word.
- Figure 1b is in my opinion inaccurate and requires correction.
Thank you for pointing this out. We think that the inaccuracy of Figure 1 is due to the conversion from Word to PDF. Because Figure 1 in the Word file is clear and has a good resolution. I will upload a new PDF file after the revision.
- I also have problems with Figure 7. I don't know if this is a manuscript processing error. Please check it
Thank you very much for catching this confusing error. It was a mistake by using the cross-reference of Figure 7 in line 218. The cross-reference referred to the entire caption of figure 7. We have now corrected this error.
- Table 1 and 4 is also split up and it's not looking good.
Thank you for this excellent observation. The reason for splitting of table 1 and 4 is due to the error that you already mentioned in comment 2. The use of the cross-reference of Figure 7 in line 218 was incorrect, which leads to recur Figure 1 three times, and split table 1 and 4 over two pages.
- Figure 11 and 12 has too little descriptions
Thank you for pointing this out. Detailed descriptions are added to the caption of figure 11 and 12.
Best regards